# A Semantic-Based Belief Network Construction Approach in IoT

**DOI:** 10.3390/s20205747

**Published:** 2020-10-10

**Authors:** Yuji Dong, Kaiyu Wan, Yong Yue

**Affiliations:** 1Department of Computer Science and Media Technology, Malmö University, 20506 Malmö, Sweden; 2Department of Computer Science and Software Engineering, Xi’an Jiaotong Liverpool University, Suzhou 215123, China; Kaiyu.Wan@xjtlu.edu.cn (K.W.); Yong.Yue@xjtlu.edu.cn (Y.Y.)

**Keywords:** uncertainty, belief, internet of things, data fusion, fault detection, self adaptation

## Abstract

Uncertainty is intrinsic in most of the complex systems, especially when the systems have to interact with the physical environment; therefore, handling uncertainty is critical in the Internet of Things (IoT). In this paper, we propose a semantic-based approach to build the belief network in IoT systems to handle the uncertainties. Semantics is the functionality description of any system component. Semantic Match mechanisms can construct the appropriate structures to compare the consistency between different sources of data based on the same functionality. In the approach, we define the belief property of every system component and develop the related algorithms to update the belief value. Furthermore, the related mechanisms and algorithms for data fusion and fault detection based on the belief property are described to explain how the approach works in the IoT systems. Several simulation experiments are used to evaluate the proposed approach, and the results indicate that the approach can work as expected. More accurate data are fused from the inaccurate devices and the fault in one node is automatically detected.

## 1. Introduction

Internet of Things (IoT) is envisioned to integrate the physical world into computer-based systems. Recently, with the advanced technology development on sensors, networking, data processing etc., IoT has demonstrated great potential in various fields [1]. However, even after decades of research on system aspects of IoT, developing IoT based systems is still facing many challenges like scalability, interoperability, and fault tolerance [2]. Because of the complex interaction with the physical environment, the IoT systems face more uncertainty than the traditional software systems and demand different fault tolerance mechanisms.

For most hybrid systems interacting with physical environments such as the outdoor nature, the hardware components can be easily affordable yet significantly affected by the nature like bad weather or wild animals, thus the systems are more unreliable due to the inaccurate detection, action or possible damage. The unreliability may cause significant loss especially in some critical systems. For example, we can use IoT systems to monitor the forests for early fire detection before it spreads. A forest fire detecting system is depending on the real-time data from the sensors deployed in the forest, however, the sensors in the outdoor environment are facing many potential uncertain damages leading to unreal detecting. The sensors can face the potential bias, drift, precision degradation, and complete failure [3]. Sometimes the sensor can also have scaled output, no sensor output or sticky sensor signal [4] from the sensor’s nature or the environmental damages. Based on the historical data from the deployed IoT system, it is common to have some unexpected behaviours from devices like the temperature sensors in California being stuck at a wrong value and remains there either permanently or intermittently [5]. There are reliable sensors and solutions; however, considering the size of the forest areas, and the number of sensors required, it is extremely difficult and expensive to maintain the systems. If we can have an affordable solution with cheap sensors while still providing reliable monitoring, it could help many large-scale IoT systems’ deployments, especially for environmental monitoring such as the early wildfire detection system to prevent large wildfires before they lose control and with a controllable budget.

The uncertainty in the hybrid system is a challenging issue due to the complex causes from different levels such as hardware faults [6], channel uncertainty in communication [7], software design [8], software development [9], and hostile attacks [10]. Especially in the large-scale and heterogeneous IoT systems, it is impossible to handle all the possible issues via any specific model.

Compared to other specifically designed models for uncertainty, the reputation-based approach is a more general solution for different cases with intrinsic self-adaptation. Since a reputation-based framework for Wireless Sensor Network (WSN) is designed in [5], this kind of approach has been widely used in the WSN applications with different technologies. In [5], each node in the WSN has two key blocks: Watchdog and Reputation. The Watchdog block is responsible for monitoring the actions of other nodes and characterizing these actions as cooperative or noncooperative, while the Reputation block is responsible for maintaining the reputation of a node based on the new observations made by the Watchdog. The reputation-based approach can usually provide better scalability and thus many different reputation-based approaches are proposed such as iterative reputation management based on belief propagation [11] and beta-based reputation evaluation system [12]. However, one rationale in the reputation-based approaches in WSN is that the sensors usually have the same functionality in the WSN applications. Therefore, they can use distance-based outlier detection [13] or density-based outlier detection [14] to evaluate other sensors’ behaviours and reputations. However, in the highly heterogeneous IoT systems, most of the devices have different functionalities making it impossible to evaluate each other’s reputation from different devices and system components.

One possible solution to deal with the heterogeneity is to integrate the social networking concept into IoT, which leads to the investigation on the Social Internet of Things (SIoT) [15]. However, defining all the relationships and behaviours among the different nodes in IoT systems from a social networking perspective remains one of the significant challenges in SIoT. To use the social networking concept for uncertainty in IoT systems, new networks with many redefined relationships and behaviours are required to meet the requirements, which is still a challenge.

In our approach, the semantics of a system component, which is the description of its functionalities, is the core concept to connect the different heterogeneous devices to build the belief network. Via extracting functionalities’ semantics, we can create some conceptual “resources” and construct a directed graph to make the system components providing the same functionality point to the same “resource” node, because they are semantically equivalent. Note that each resource only stands for one functionality from the different components and one component may point to multiple different resources. For example, an underwater robot may monitor the fish density and water quality with different parameters [16], and the robot can point to different resources based on its own functionalities with monitoring different parameters. Since the system components that point to the same “resource” have the same type of functionality, they can evaluate each others’ behaviours and thus give a belief value. We assume the majority of the components in the system are running as normal, and the fault appears rarely. If a few components in the IoT system provide the same functionality in the context, they will always have similar data when they are running as normal. Whenever the data produced from one component are so different from the others, we assume the component has a higher risk to have fault than the other components with the consistent behaviour. Figure 1 indicates this assumption that once a component produces a very different value compared to other components with the same functionality, we think the component is in an abnormal state. One component may have multiple different functionalities. We assume the component has a higher risk to have faults once its one functionality has abnormal behaviours. The whole belief updating mechanism is based on the above assumptions. If most of the system components can run normally and the faults only happen rarely, we can use the normal running components to evaluate all the components’ belief values to detect some abnormal behaviours based on the components with the same “semantic”. The belief-based approach proposed in this paper is a general and distributed method that can be seamlessly integrated into any IoT systems.

The proposed approach can use semantic to build a belief network in the heterogeneous IoT systems, which can be used for data fusion and self-adaptive fault detection. It is an approach extremely useful for the large-scale IoT system deployment in the outdoor environments, because the outdoor environment face many potential risks for the devices such as the bad weather or wild animals, and the approach can provide flexible, low-cost solutions with cheap, inaccurate, and unreliable devices to build a more accurate and robust system.

This paper is organised as follows; in Section 2, we present some related works on data fusion and fault detection with different approaches, especially reputation-based methods in IoT or WSN. Our Semantic-based Belief Network Construction Approach is described in Section 3 which is divided into four parts. Section 4 introduces how this approach can be implemented in an IoT simulator. The evaluation results are expressed and discussed in Section 5. Finally, Section 6 concludes the paper and discusses the future work.

## 2. Related Works

The issues of uncertainty in IoT systems come from many different aspects such as physical randomness, noise, software faults, or attack. The research fields and technologies in handling uncertainty in the hybrid systems mainly include data fusion, fault diagnosis, security, etc. In this paper, the proposed approach focuses on the data integrity and accuracy with data fusion, and adaptive fault detection.

Ever since the multi-sensor data fusion was developed, it benefits many different fields like military, manufacturing and robotics. The utilisation of data fusion in the different system requires an understanding of basic terminology, data fusion processing models, and architectures [17]. In general, the approaches for data fusion can be classified into two types, model-driven and data-driven. The model-driven approach needs prior knowledge of the expected data model and can be more domain specific, while the data-driven approach has a more general principle and becomes more popular for sensor networks in the recent years.

A general reputation framework is proposed in [5] for high integrity sensor networks with peer-to-peer rating reputation based on sensor nodes’ behaviours via the watchdog components. Since the data authentication problem in WSN is modelled as a problem of developing a community of trustworthy sensor nodes, the proposed approach is a general and unified solution and can be easily integrated in existing deployments without making many modifications. However, the approach only employs Bayesian formulation to manage the reputations, which may not always be the best algorithm in the given context. Based on different requirements in the different application domains, many different algorithms like probability theory, fuzzy set theory, and the Dempster–Shafer evidence theory are used for reputation management in the multi-sensor data fusion, and they all have their own advantages and limitations. For example, an evidential approach based on the Dempster–Shafer evidence theory can allow uncertain and ambiguous data for data fusion; however, the fusion becomes more imprecision and inefficient with highly conflicting data [18]. Many modified algorithms are proposed to overcome the different limitations. For example, Ref [19] proposed a method based on the cosine theorem to measure the conflicts between different evidence in the data fusion based on the Dempster–Shafer evidence theory. In [20], Deng entropy is adopted to measure the uncertain information and evidence distance is applied to measure the conflict degree. They can handle the conflicting evidences better in the data fusion based on Dempster–Shafer evidence theory in WSN. However, not only the different algorithms can affect the data fusion results, the mechanisms of how to observe nodes’ behaviours could also change the system performance.

Different from some approaches simply applying fusion algorithm among neighbour nodes, Ref [21] used a variety of weighted trust factors and coefficients related to the systems to obtain direct and indirect trust values. The approach considered more factors in evaluating the reputation of the different nodes, which successfully underlines the fuzziness, subjectivity and usability of trust. However, a lot of direct and indirect trust evaluations with cooperation and communication with neighbors may need excess energy and time costs. The exponential-based trust and reputation evaluation system (ETRES) [22] used exponential distribution to represent the distribution of nodes’ trust in WSN and utilise the entropy theory to measure the uncertainty of direct trust values. The indirect trust is only used to strength interaction information when the uncertainty of direct trust is too high. Since the indirect trust is not always required, the approach can reduce the computing power of nodes and prolong the lifetime of the network. However, the system performance is limited with ETRES, which may decrease the security level. For large-scale wireless sensor networks, Ref [23] proposed a novel trust estimation approach with clustering to improve cooperation, trustworthiness, and security with less memory and power consumption. The design of two levels of clusters makes the minimum overheads. The trust decision-making schemes at the different clustering levels can improve resource efficiency and cooperation among different clusters. However, the proposed work did not provide memory overhead and was not suitable for big misbehavior. For underwater wireless sensor networks, Ref [24] proposed a Hidden Markov Model (HMM) based trust management model for measuring the trustworthiness of the sensor nodes based on their dynamic behaviors. The trust is evaluated by observing the interaction quality between one node and its neighbours. The HMM algorithm is a good tool for sequential behaviors and thus could be a good choice in the dynamic environment. However, the assumptions of malicious nodes are very simple and may not cover many potentially malicious nodes.

Even though all the above reputation-based approaches have their advantages in some domains, all similar approaches are mostly designed for homogeneous WSN systems, where every sensor node provides the same functionality. A sensor node can only evaluate another sensor nodes’ behaviours when the node can perform a similar behaviour and it is a foundation for many reputation-based approaches in WSN for data fusion and fault detection. For example, a temperature sensor cannot evaluate the behaviours from another humidity sensor, because the node cannot know the correct behaviours from a heterogeneous device.

Even though many trust evaluation methods have already been proposed for different kinds of WSN, those existing solutions cannot be directly applied to highly heterogeneous systems like IoT, because the devices all have different functionalities. It is extremely difficult to observe the nodes’ behaviours via other nodes as their neighbour nodes may have a completely different functionality and behaviour. Furthermore, besides sensors, there are many other types of system components in IoT. This reveals difficulty in integrating those components into the old reputation-based approaches provided for WSN. Some researchers use the social networking concept in the IoT systems to build more complex networks with social relationships to construct the reputation framework. In [15], a solution which used to be used in P2P and social networks is proposed for trustworthiness management in the IoT field. The trustworthiness of a friend for one IoT node is evaluated based on the nodes’ own experience and is the opinion of the friends in common with the potential service providers. The author proposed a subjective approach which can deal with the nodes with dynamic behaviors. However, a subjective approach will have a slower transitory response and may become difficult in a complex network. From implementation perspective, a dynamic trust management protocol is proposed in [25] for IoT to deal with misbehaving nodes whose status or behavior may change dynamically. The trust of each IoT node is evaluated based on the social relationships between different IoT nodes with three trust properties, honesty, cooperativeness, and community-interest. The proposed approach can make the trust evaluation converges to the ground truth status in dynamic IoT environments and be resilient to misbehaving attacks. However, the evaluation only considered the increasing hostility as an instance of changing environment conditions, which is probably different in a more realistic environment. In [26], the author proposed a scalable, adaptive and survivable trust management for dynamic IoT environments based on a community of interest (CoI), where the nodes can dynamically join and leave based on their communities of interest. The newly joined node can quickly build up its trust relationship with desirable convergence and accuracy behavior based on existing trust information in the network. The scalability is achieved from a storage management strategy that each node only needs to keep the trust information towards a subset of nodes based on its interest and storage space. However, the concept of CoI may act differently in the different application domains, and thus need more investigations in the different contexts.

The domain of Social Internet of Things (SIoT) is a cutting-edge research field and has many challenges such as scalability, lookup, communication protocols, and social networking management [27,28]. Because of the different foundations of social networks and IoT systems, many concepts have to be redefined from a new perspective. For example, “Honesty” is natural concept in social networks, however, it does not have a clear meaning in the IoT field for an IoT node. In [29], the author defined some trust attributes that directly affect to honesty and proposed a subjective computational model to evaluate the honesty based on experiences of objects and opinions from friendly objects with respect to identified attributes. The defined honesty can be used to evaluate the trust from a different dimension in the IoT. However, it was not clear about how to use the honesty concept in the real IoT systems and it needed more investigations. In [30], the trust concept, definition, and model are rebuilt in the context of the SIoT to support the trust evaluation in the SIoT environment. The author proposed a trust evaluation model mainly from three perspectives: Reputation, Experience and Knowledge. The proposed trust model provided an interesting insight about the different levels of trust in the SIoT context. However, it may need more case studies to explore the concept in the real IoT systems. In [31], a reputation oriented trust model called StoRM is proposed for IoT by combining social dimensions and microservice architecture with agent technology. The agents acted in the environment as a social network, and thus they can propagate the requests to the rest of the agent community like its neighbours to evaluate the reputation based on some social principles. However, how the agent finds the appropriate neighbours for reputation evaluation may become a problem in the real IoT systems. To sum up, the SIoT is an interesting direction for trust management in IoT, however, the solution could be complex since the approaches used in social networks are not necessarily compatible with IoT. Many concepts and solutions may need to be redefined and redesigned to fit the IoT domain.

Different from the works mentioned above, the Semantic-based Belief Network Construction Approach (SBNCA) proposed in this paper is a general and flexible solution for different scenarios and the solution is built based on the semantics of different system components instead of social network structure. Semantics is an important high-level concept which is used in many IoT challenges. For example, in the [32], the semantics is used in the IoT framework to support RESTful devices’ API interoperability. Based on the sensors’ APIs RESTful descriptions and the interfaces exposed by smart sensors, a common Aggregator can integrate different sensors’ interfaces. The semantic is the functionality provided by the sensor and it can provide high level abstraction. In this paper, the approach can match the different system components with different functionalities and behaviours based on high-level semantics to construct a belief network. The belief network can reduce the runtime uncertainty and self-adaptively catch some general fault. Therefore, the SBNCA can not only deal with highly heterogeneous IoT systems, but also provide the self-adaptation feature in data fusion and fault detection in the large-scale IoT systems.

## 3. SBNCA (Semantic-Based Belief Network Construction Approach)

The Semantic-based Belief Network Construction Approach can be used to build the belief network in the IoT systems and it can be divided into four parts: semantic match, data fusion, belief updating and fault detection. In particular for device noise, the data fusion is used to support more reliable and accurate aggregated values from the different devices, and for unpredictable fault, the fault detection can self-adaptively detect the unknown faults. The detailed uncertainty models are discussed in the respective section. The semantic match is to construct the special structures for the heterogeneity in the IoT systems, and the belief updating provides the mechanism for evaluating the belief values, which stands for how much you can trust the system component. Figure 2 visualises the overall concept of this approach where the green nodes are IoT system’s components and the blue nodes are conceptual “resources” representing the components’ semantic which are the description of their functionalities. At the moment captured by Figure 2, three system components have abnormal running statuses which are marked as red nodes. The overall directed graph structure is the semantic match. The blue nodes of “resources” receive data from the green nodes of “components” to implement data fusion and belief updating. When some nodes’ belief values are too low, the fault detection is triggered and the nodes are marked as red. Before the detailed structures and algorithms are presented, several important concepts are defined as follows:

**Definition** **1**(Uncertainty)**.**
*For the uncertainty in the IoT systems, the Semantic-based Belief Network Construction Approach is mainly designed for two types of uncertainty—device noise and unpredictable fault in the devices. The typical devices used in the IoT, like sensors, always have the deviation of measurements. Based on the central limit theorem [33], even though the individual constituent deviations may not be Gaussian distributed, the combined deviation is approximately so, therefore we assume the uncertainty from the device noise follows the Gaussian Distribution. For the unpredictable fault in the device, it is defined as the general fault caused from any possible reasons. For example, the device may be damaged by animals, and starts to give wrong values. The general faults in the devices can be detected by the fault detection part in SBNCA.*

**Definition** **2**(Component)**.**
*The component is defined as the system component, containing both the software and hardware parts. The component is the entity which has a series of functionalities in the IoT system. For example, it could be a humidity sensor to detect humidity in the air.*

**Definition** **3**(Resource)**.**
*The concept resource is a conceptual mapping to a set of entities, which was originally proposed in the REST (Representational State Transfer) architectural style to build the modern web [34]. The definition of the resource is narrowed down in the Semantic-based Belief Network Construction Approach, where the resource can be treated as the functional description of the components from high-level semantics. If the system has one temperature sensor at location $l_{\left(x,y\right)}$ to detect temperature T and the function can be written as $T_{l_{\left(x,y\right)}}\left(t\right)$ where the t is the time variable, the function $T_{l_{\left(x,y\right)}}\left(t\right)$ can be a resource mapping to “the real-time temperature at location $l_{\left(x,y\right)}$” with a unique URI (Uniform Resource Identifier) to name and address this resource. Any resource can be an abstract concept or function mapping to a type of data that can be detected or evaluated by the components in the IoT systems.*

**Definition** **4**(Belief)**.**
*Belief is the value to describe the reputation of the component. The value of the belief is given by the mapped resource to express how much the component can be trusted.*

Figure 2 illustrates a big picture of the runtime framework in which many different system components are matched to different resource nodes with similar semantic functionalities. The blue nodes are resources expressing the functional semantics matched from the components. The arrows from different devices to the resources indicate the semantic match relationships. The resource nodes will fuse the data from different components and give the evaluated belief properties to all the matching components. The system will detect the fault if any components’ belief property is lower than the given thresholds. In the figure, the green nodes are running as normal, while the red nodes are in the detected fault mode.

### 3.1. Semantic Match

Reputation-based approaches have been widely used in many WSN applications to deal with uncertainty. However, it is very challenging to apply reputation concept in the IoT systems because most system components are heterogeneous and multi-functional, therefore it is difficult to require any system component to evaluate other system components’ behaviours when they have entirely different functionalities.

However, even in the most heterogeneous systems, some system components still have the same or similar functionalities. If we can construct a special structure to check the data and behaviours between these components, it is possible to have distributed reputation evaluating processes via consensus-checking between different system components which are semantically equivalent. To achieve this goal, in this framework, we use a commonly utilised abstract concept in the web resource, to build the consistency-based structure for evaluating the reputations, and the process of building the structure is named as semantic match.

After a conceptual resource is created, several different system components can be bound to this resource with some specific functionalities matching to the resource. A classical example to explain this concept is the hybrid localisation solution [35] in which the GPS positioning component, WiFi positioning component and Cellular positioning component all map to a concept of the entity location as three semantically equivalent entities. Thus we can create a resource $e_{\left(x,y\right)}$ to express the location of entity *e* and bind the three components to this resource. This solution cannot only be used to make the localisation more accurate and reliable but also make it possible to detect any abnormal fault from these three components, illustrated in Figure 3. Because in the long-run, if any component suddenly starts to give unbelievable drift values compared with the other two components, it is possible that the component may have deviated from the normal running status.

A complex IoT system may contain many different components in addition to a few localisation modules, thus the system can construct a directed graph containing the system component nodes, the conceptual resource nodes, and the arrows pointing from the system component nodes to the conceptual resource nodes. As an example, Figure 4 illustrates a directed graph with six individual components and four resources.

To give a formal description, any arrow $a_{ij}$ from $c_{i}$ to $r_{j}$ indicates a functionality of the component $c_{i}$ as a sequence of data with variable time *t*
(1)$$f_{a_{ij}}\left(t\right)=\left({\vec{d}}_{c_{i},r_{j},t_{1}},{\vec{d}}_{c_{i},r_{j},t_{2}},\;\ldots ,\;{\vec{d}}_{c_{i},r_{j},t_{k}}\right)$$
where any ${\vec{d}}_{c_{i},r_{j},t_{p}}=<v_{1},v_{2},\;\ldots ,\;v_{m}>$ is a vector produced from component $c_{i}$ at time $t_{p}$. Any resource $r_{\mu }$ has a unique $URI_{r_{\mu }}$. At time $t_{p}$, it receives α number of data from α number of system components and the received data in resource $r_{\mu }$ is expressed as a set of ${\vec{D}}_{r,t_{p}}$:(2)$${\vec{D}}_{r_{\mu },t_{p}}=\left\{{\vec{d}}_{c_{1},r_{\mu },t_{p}},{\vec{d}}_{c_{2},r_{\mu },t_{p}},\;\ldots ,\;{\vec{d}}_{c_{\alpha },r_{\mu },t_{p}}\right\}$$

Generally, any system component $c_{i}$ can have several different functionalities to be mapped to different conceptual resources. Each functionality can produce a vector of data ${\vec{d}}_{c_{i},r,t_{p}}$ at time $t_{p}$ and the data will be sent to the resource *r*. Furthermore, each system component $c_{i}$ has a special property $b_{c_{i}}$ as the Belief of the component $c_{i}$ assigned from the matching resources. Because every system component has its mapping resources, the mapping resources can keep the Belief property updated. Essentially, the process of the semantic match is to add some extra resource nodes in the resource registry and construct the specific structures with algorithms to check consistency between different components with the same functionality. After the semantic match structure is built, the resources can fuse data from different components to produce more accurate results and use the results to update all matched components’ Belief property to evaluate their reputations.

### 3.2. Data Fusion

Because one resource has different sources of data from different components, it is possible to integrate different data to achieve more accurate and reliable data. This process is called data fusion. The targeted uncertainty is from the device noise, and it is assumed that the accuracy of the devices follows the Gaussian Distribution. Based on the redundant data from the different components, the effects of the device noise can be reduced by applying the algorithms. There are many different approaches for the data fusion, especially when there is real-time updated Belief property for each source. In this paper, for simplicity, we only use a linear model based on Fuzzy Set [36] to explain how the data fusion works in our framework.

The belief $b_{c_{i}}$ of any component $c_{i}$ is between 0 and 1, $0<b_{c_{i}}<1$, that $b_{c_{i}}=1$ means the system believes the component $c_{i}$ is 100% correct and $b_{c_{i}}=0$ means the system believes the component is 100% wrong.

A component has its belief property $b_{c_{i}}$ and the output from this component to a resource *r* at time $t_{p}$ is a vector ${\vec{d}}_{c_{i},r,t_{p}}=<v_{i},v_{2},\;\ldots ,\;v_{m}>$. Due to the device noise uncertainty, we assume the real data are ${\vec{d}}_{c_{i},r,t_{p},real}$ and $b_{c_{i}}\cdot {\vec{d}}_{c_{i},r,t_{p}}\leq {\vec{d}}_{c_{i},r,t_{p},real}\leq \left(2-b_{c_{i}}\right)\cdot {\vec{d}}_{c_{i},r,t_{p}}$, thus:(3)$${\vec{d}}_{c_{i},r,t_{p},real}=<v_{1,real},v_{2,real},\;\ldots ,\;v_{\varrho ,real},\;\ldots ,\;v_{m,real}>$$
where 1≤ϱ≤m and $\forall v_{\varrho ,real}\in \left[b_{c_{i}}\cdot v_{\varrho },\left(2-b_{c_{i}}\right)\cdot v_{\varrho }\right]$. We use:(4)$${\vec{d}}_{c_{i},r,t_{p},real}=\zeta \left(b_{c_{i}},{\vec{d}}_{c_{i},r,t_{p}}\right)$$
to express the real data at time $t_{p}$ from the component $c_{i}$ to the resource *r*.

Any resource *r* is mapped from a set of components $C_{r}$, where any component $c_{i}\in C_{r}$ produces the vector data ${\vec{d}}_{c_{i},r,t_{p}}$ at time $t_{p}$. Let us assume the resource *r* successfully received correct time-stamped data from all the source components in $C_{r}$ and the size of the components is $\kappa =|C_{r}|$. The received data in the resource *r* can be denoted as a set of vector data ${\vec{D}}_{r,t_{p}}$:(5)$${\vec{D}}_{r,t_{p}}=\left\{{\vec{d}}_{c_{1},r,t_{p}},{\vec{d}}_{c_{2},r,t_{p}},\;\ldots ,\;{\vec{d}}_{c_{\kappa },r,t_{p}}\right\}$$
and the respective belief values from all the matched components can be expressed as a vector $b_{r,t_{p}}=<b_{c_{1},t_{p}},b_{c_{2},t_{p}},\;\ldots ,\;b_{c_{\kappa },t_{p}}>$. Then, based on the Definition Equation 4, the real value should be expressed as:(6)$${\vec{D}}_{r_{real},t_{p}}=\left\{\zeta \left(b_{c_{1},t_{p}},{\vec{d}}_{c_{1},r,t_{p}}\right),\zeta \left(b_{c_{2},t_{p}},{\vec{d}}_{c_{2},r,t_{p}}\right),\;\ldots ,\;\zeta \left(b_{c_{\kappa },t_{p}},{\vec{d}}_{c_{\kappa },r,t_{p}}\right)\right\}$$

Since we eventually need a single scalar quantity output for further services and to update the belief properties, the defuzzification is required to produce the expected output. There are many different methods to do the defuzzification such as max membership principle [37], centroid method [38], weighted average method [39], or mean max membership [40]. In this paper, we assume the uncertainty of the devices fitting Gaussian Distribution, therefore, the output membership function of ${\vec{d}}_{c_{i},r,t_{p},real}=\zeta \left(b_{c_{i}},{\vec{d}}_{c_{i},r,t_{p}}\right)$ is symmetrical. It satisfies the condition of the weighted average method, which is more computationally efficient, so we choose it to produce the results. The general algebraic expression of the weighted average method is defined in [37] as follows:(7)$$z^{*}=\frac{\sum \mu_{\tilde{c}}\left(\bar{z}\right)\cdot \bar{z}}{\sum \mu_{\tilde{c}}\left(\bar{z}\right)}$$
where ∑ denotes the algebraic sum and $\bar{z}$ is the centroid of each symmetric membership function. In our framework, the expected real output value can be expressed as follows:(8)$${\vec{d}}_{t_{p}}^{*}=\frac{\sum_{i=1}^{n}{\vec{d}}_{c_{i},r,t_{p}}\cdot b_{c_{i}}}{\sum_{i=1}^{n}b_{c_{i}}}$$

The data fusion process is running at the resource $r_{\mu }$ with $URI_{r_{u}}$, which expresses the semantics of $r_{\mu }$. The resource $r_{\mu }$ collects all the incoming data from the semantically matched components with the same time stamp. Combined with all the related belief values from different components, the resource $r_{\mu }$ can calculate the fusion data ${\vec{d}}_{t_{p}}^{*}$ based on Equation (Equation 8) as expressed in Figure 5. The calculated data ${\vec{d}}_{t_{p}}^{*}$ are attached to the resource $r_{\mu }$ to express the fusion result, which could be treated as the semantic value. The fusion result ${\vec{d}}_{t_{p}}^{*}$ can also be used in the next belief updating part to update all the matched components’ belief values.

To take an example showing how the data fusion works, we assume the framework contains three positioning components; $c_{1}$, $c_{2}$ and $c_{3}$ with respective belief value of 0.9, 0.6 and 0.4. At the time *t*, if the three positioning components respectively send their detected location values <56,43>, <63,47> and <40,35> to the resource $r_{p}$ with $URL_{r_{p}}:systemX/entityA/location$. The evaluated result ${\vec{d}}_{r_{p},t_{p}}^{*}$ in the resource $r_{p}$ is:$$\begin{array}{cc} {\vec{d}}_{r_{p},t_{p}}^{*} & =<\frac{\sum_{i=1}^{n=3}{\vec{d}}_{c_{i},r_{p},t_{1}}\cdot b_{c_{i}}}{\sum_{i=1}^{n=3}b_{c_{i}}},\frac{\sum_{i=1}^{n=3}{\vec{d}}_{c_{i},r_{p},t_{2}}\cdot b_{c_{i}}}{\sum_{i=1}^{n=3}b_{c_{i}}}>\\ \; & =<\frac{56\cdot 0.9+63\cdot 0.6+40\cdot 0.4}{0.9+0.6+0.4},\frac{43\cdot 0.9+47\cdot 0.6+35\cdot 0.4}{0.9+0.6+0.4}>\\ \; & =<54.8,42.6> \end{array}$$

The value of ${\vec{d}}_{r_{p},t_{p}}^{*}$ is the semantic value of $r_{p}$ to express the location of entityA.

### 3.3. Belief Updating

To obtain the correct evaluated data based on the above data fusion process, one of the most important factors is to assign the correct Belief values to all the components.

There are many different methods to assign the components’ Belief properties. The method can be static, based on the historical data and prior knowledge. It can also be dynamic, based on Bayesian probability or other theories. In this Section, we only give some requirements as the guide to use this framework, in which the algorithms updating the belief can be flexible.

Different from other approaches, the belief updating processes in our approach are not individual end-to-end processes from different nodes rating each other. Each resource can become an individual agent to decide all the matched system components’ beliefs based on their behaviours, which is a centralised decision in the resource rather than peer-to-peer decision.

Based on all the data from different component sources in the same semantic match, the resource can update its mapped components’ Belief properties. The updating rules are based on the following two basic assumptions:If more resources think one component is correct, the component has a higher reputation.If the outputs from different components are more consistent, these components are more believable than others and will get a higher reputation.

With the two assumptions, we can have the belief updating mechanism in the following:

For any component $c\in C_{r_{i}}$, where $C_{r_{i}}$ is the set of all semantically equivalent components to the resource $r_{i}$, we have the Belief of this *c* as:(9)$$b_{c_{r_{i}}}=\hbar \left(c,C_{r_{i}}\right)$$
where the $\hbar (c,C_{r_{i}})$ is a function to check the difference between the data from the component *c* and other components in the $C_{r_{i}}$. The function will get a higher value if the difference is less, and a lower value if the difference is more. The equation means that if the component’s output values are more close to other components’ that are semantically equivalent to this resource, the component’s output values are assumed to be more accurate.

Many system components are multi-functional, which means they can be mapped to different resources based on the semantic equivalences between the functionalities and resources. If a component *c* is mapping to multiple resources $R_{c}$, every mapping resource can give the belief values to it separately, and eventually the Belief of this component *c* is:(10)$$b_{c}=g\left(B_{R_{c}}\right),B_{R_{c}}=\left\{b_{c_{r_{i}}}|r_{i}\in R_{c}\right\}$$
where the $g(B_{R_{c}})$ is a function used to integrate the beliefs from different resources and the range is between 0 to 1. In general, this equation can be summarised in the following: if more children components are reliable, the parent component is believed to be more reliable.

The above two functions only indicate the underlying assumptions, however, the exact algorithms need to consider many different aspects such as the network latency and data loss. In the Section 4, we will illustrate how to update the belief for each different component using our data structures.

### 3.4. Fault Detection

With the real-time updating Belief properties, any system component in the system can self-adaptively detect some runtime faults, especially when one system component causes some permanent faults.

We can set a threshold value Ψ, and if any component has the belief value lower than Ψ, the system will trigger the event of warning the related components. In some complicated situations, the different components can also have different threshold values, thus we can have more fine-grained operations on the fault detecting.

The fault detection is based on components’ belief values, which are automatically updating at runtime, therefore the fault alarm is self-adaptive, which can let the deployed IoT systems to detect any unexpected fault in real-time without any prior knowledge. This is important because it is impossible to consider all possible states when developing the software systems, and this self-adaptive fault alarm mechanism can deal with most of the undefined behaviours and data.

## 4. Implementation

The implementation is based on an open-source IoT simulator IoTNetSim [41], which could provide end-to-end IoT services and networking. We extended the IoTNetSim to support SBNCA and the source code is open at github (https://github.com/yujidong/Semantic-based-Belief-Network-Construction-Approach). The SBNCA can be divided into four parts: semantic match, data fusion, belief updating and fault detection based on Section 3 and we will express how the approach can be implemented and simulated in IoTNetSim in this Section.

The most significant part in SBNCA is semantic match, where we can build the fundamental structure to utilise the approach with other mechanisms. The principle in semantic match is to match the different components in IoT systems when they have the same semantic, which can be also explained as their functionality description. To simplify the implementation, we give a determinate semantic description in every sensor node in our implementation to describe the functionality of the sensor node. However, it is possible to use rich semantics related technologies like Resource Description Framework (RDF) with logic-based technologies to do reasoning and inferring in our approach in order to express more complex semantics, thus some existing artificial intelligence technology in semantic reasoning with uncertainty like [42,43] can be used in the IoT systems with our proposed approach. The implemented semantic contain some important functional information such as the sensor type and location. Each sensor node will have a responsible gateway node for further processes and the responsible gateway node will act as the resource node. If the different sensor nodes managed by the same gateway node has the same semantic, they can build a semantic match structure as shown in Figure 6.

Figure 6 is an abstract data structure and the implementation in IoTNetSim is based on a HashMap in the gateway node. The gateway will act as an agent to decide if the upcoming message from a sensor has the same semantic with any other connected sensor. If there is an existing same semantic in the resource (gateway) node, the resource node will add the sensor node in the semantic match structure to continue data fusion, belief updating and fault detection.

The existing implementation of semantic match can support the location-aware scenarios, where a semantic from a sensor can be expressed as the “perspective functionality *f* at location *l*”. Even though the existing semantic expression is extremely limited, the approach already can be useful in some scenarios. For example, in the scenarios of environmental monitoring for pollution detection or forest fire detection, the large-scale IoT deployment is always a tricky issue. Because the comprehensive environmental monitoring requires enormous numbers of sensors to cover all the area and the complex outdoor environments can make such systems extremely unreliable, it is always difficult to make the decisions between the cost and reliability. However, with the support from SBNCA, we can deploy a large number of cheap and unreliable sensors in the natural and let a drone or robot with more reliable and accurate sensors to patrol in the area. As shown in Figure 7, the given outdoor area can be divided based on the grid and the measure metrics in different sub-area can be expressed as a specific semantic. For example, a semantic can be expressed as the “temperature at location $l_{i}$”, which is not tightly attached with any sensor. Whenever a drone is passing by the area at location $l_{i}$, if the drone has a temperature sensor, it can join the semantic match structure and contribute its own detection. Since the drone should have much better care than the outdoor sensors, we have much more flexible choices for the large-scale IoT system deployment with both cost and reliability concerns. Even though the deployed static sensor nodes are all cheap, inaccurate and unreliable, we can still deploy some movable nodes with more expensive, accurate and reliable devices to help the static nodes. The overall system can have a better performance from all the nodes with different belief values.

After the semantic match structure constructed, the general process in the resource node is expressed in Algorithm 1. Every semantically matching sensor can send the request to the related resource nodes to fuse their data and get updated belief value. The implemented data fusion algorithm is based on the FuzzyWeightedAverage algorithm introduced in Section 3.2.
**Algorithm 1:** The Process from a Resource Node to Implement SBNCA.
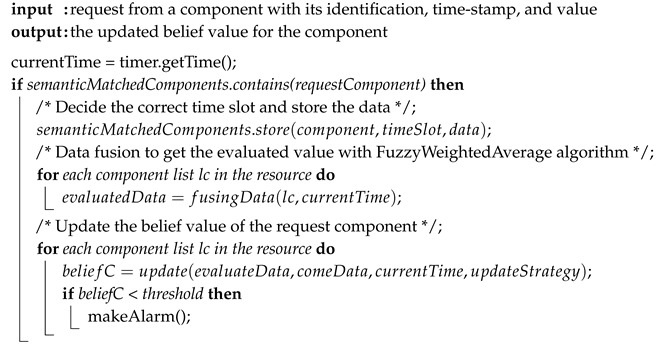


The belief updating strategy is based on both historical belief values and the current calculated value, therefore another buffer is used to store historical belief values. The historical belief values can be used to apply PI (proportional-integral) control mechanism [44] to stabilise the belief value. In the implementation, we can set a length of *l*, and only consider the data of the latest *l* number of belief values. The length of *l* is critical because it can significantly affect the performance of the approach. Generally, if the length *l* is longer, the evaluated belief value will be more stable and fault-tolerant. However, it will also become very insensitive to all faults in the meantime, thus some faults may not be reported promptly. Therefore, the parameter needs to be chosen carefully to balance the different requirements from the specific systems.

The data fusion algorithm is not limited in SBNCA and so is the belief updating strategies. Different algorithms may have different performance in the different scenarios, which is depending on the detailed system requirements. The implementation here is just to give an insight of how the approach works and we will indicate more evaluation results in Section 5.

## 5. Evaluation

The data originally provided in the IoTNetSim does not fit the requirements of SBNCA, because the data are only collected every two hours, where the interval is too big to make SBNCA work. In order to provide more realistic and continuous data for evaluation, we use the real sensor data which are collected every 5 min from the City of Melbourne [45].

As shown in Figure 7, we assume the simulated IoT system is located in a large area, which can be divided by a grid structure. The system intends to monitor the environment for further purposes such as the pollution management or early forest fire detection. Since we need a large number of sensors to cover all the area, the sensors have to be cheap, inaccurate and unreliable. The IoTNetSim can easily simulate a large number of sensors, however, because the area can be divided by the grid and each sensor only need to be responsible for the limited given area, the scale of the system will not affect the evaluation results much. The following simulation is based on the real data from [45] and the sensor types are limited. However, since the data fusion and belief updating are only depending on the data produced from the devices while not related to the detailed functionalities of the devices. A more complex IoT system will only make the semantic match part more complex, while the data fusion, belief updating, and fault detection parts within a semantic match will keep the same as following experiments. Therefore we only used two types of sensors and one movable drone carrying the sensors to explain how the proposed SBNC can be deployed in the real environment. We can add more sensors types, robots and drones, however, it will not affect the results.

In the simulation, we divide the area into nine locations, where each location has a link node and gateway node. In each location, we have three temperature sensors and three humidity sensors. Apart from the static sensors, we have one moving temperature sensor and one moving humidity sensor. The moving sensors are assumed to be with a drone, which can patrol in the area. All the static sensors are assumed to be cheap, inaccurate and unreliable, while the sensors on the drone are expensive, accurate and reliable. Whenever the drone moves to a different area, it will check the local link node and gateway node. If it finds the matchable semantic, it will join the semantic match structure with other static sensors. When the drone is patrolling in the area, it acts as a validation node, which can help the data fusion and belief updating for all the other nodes. Figure 8 indicates how the drone can join different semantic matches in the different areas, so the proposed approach allow the drone to carry more expensive sensors to provide better data fusion and belief updating for all the nodes in the IoT system. Since it is a validation node, its belief value will be fixed in a higher level instead of estimated by other nodes.

Since the sensors on the drone are acting as the validation sensors, their belief values are almost 100%. We give them 99% in the experiment and they will not be updated by any other node, because different from other static sensors, the sensors on the drone can be maintained regularly with low cost and thus be more trustable.

In SBNCA, the belief updating part only gives the basic assumptions without detailed algorithms. In the experiments, we use a simple linear algorithm to evaluate the belief values. That is, if the evaluated data are $d_{e}$, and the requested data are $d_{r}$, the belief value for component *i* is:(11)$$b_{i}=1-\frac{|d_{e}-d_{r}|}{d_{e}}$$

Since the belief value should be an essential feature of the component, it should not change too much in every second. Unreliable components can also accidentally produce the correct results sometimes, however, the belief value should be more statistically significant.

To give better belief evaluation, we need to stabilise the belief value, so the evaluation depends not only the current data, but also the historical data. In this paper, we use the widely used discrete PI (Proportional Integral) control mechanism. The mathematical form of discrete PI control function can be expressed as:(12)$$u\left(t\right)=K_{c}\;e\left(t\right)+\frac{K_{c}}{\tau_{I}}\sum_{i=1}^{n_{t}}e_{i}\left(t\right)\Delta t$$
(13)e(t)=SP−PV
where the two tuning values for a PI controller are the controller gain, $K_{c}$ and the integral time constant $\tau_{I}$. The value of $K_{c}$ is a multiplier on the proportional error and integral term and a higher value makes the controller more aggressive at responding to errors away from the set point. The set point (SP) is the target value and process variable (PV) is the measured value that may deviate from the desired value. The error from the set point is the difference between the SP and PV and is defined as e(t)=SP−PV. Since the controller is discrete, the classical continuous form of the integral is changed to a summation of the error and use $n_{t}$ as the number of sampling instances.

Since the historical data are already stored in the semantic match structure as shown in Figure 6, we design following equation to calculate the belief value of component *i* based on Equations (Equation 11)–(Equation 13):(14)$$b_{i}\left(t\right)=\gamma \cdot {b_{i}}^{n_{t}}+\frac{1-\gamma -\delta }{l}\cdot \sum_{j=n_{t}-l}^{n_{t}-1}b_{i}^{j}+\delta \cdot {b_{i}}^{0}$$

The parameter *l* is the size of the timeslots from the latest historical belief values and $b_{i}^{n_{t}}$ stands for the belief value at the timeslot $n_{t}$. We use Equation (Equation 14) to calculate the belief value of each component. In the following experiments, we set the variable γ as 0.3 and δ as 0.4. It means that the belief evaluation of the component *i* has 40% impact from the device nature, 30% impact from the current performance and 40% impact from the latest performance in a given period. The concern is from the sources of the uncertainty.

Since the entire area is divided into many different sub-area based on a grid structure and the sensors have similar performance for different sub-areas, we only choose one sub-area to show the performance in the following experiments. To be noticed, all the experiments are based on nine locations with 56 sensors and the scale of the system can still increase without changing the system performance. However, all the discussions in the following experiments are based on the analysis of the system components with a same semantic. They are the temperature sensors at location 1.

In the first experiment, we initialise all the static sensors’ belief value to be 60%, which means all the static sensors are extremely inaccurate and unreliable. Since the entire area is divided by the grid, the system performances in the different locations are similar. We choose the three static temperature sensors in location 1 to show the experiment results as shown in Figure 9. As the sensors’ initial belief value is only 60%, we can clearly see that their detected temperature values are usually far from the real value in the red line in Figure 9a. However, after the data fusion, the fused data in the black line in Figure 9a are much more accurate and reliable. The average deviation of the evaluated data and the real data in this experiment is only 12%, while each static sensor should have 40% error rate based on the setting. It proves the effectiveness of the data fusion in SBNCA that it can produce more accurate data based on multiple inaccurate devices. In Figure 9b, we can see the belief values of all the 3 sensors are mostly changing between 50% and 70%, which shows the good belief evaluation for the sensors.

In the second experiment, we set the different accurate levels for different sensors. The three static temperature sensors indicated in Figure 10 have 85%, 70% and 55% accurate rate, respectively. The performance is as good as the first experiment, where the average deviation of the evaluated data and the real data are only 10%. In the belief updating part, all the sensors’ belief values are also properly evaluated and it is a self-adaptive process. Since the system can self-adaptively fuse the data and update belief values as shown in the experiment results, we can infer that the approach can also work in a large-scale environment.

The third experiment is to validate the fault detection in SBNCA. In the experiment, the sensor3 will only detect 0 after running for a while and the other two sensor nodes are still working properly. As explained in Section 3.4, the fault detection method is to set a threshold value Ψ to detect any component which has its belief value lower than Ψ. The Ψ in this experiment is 40%. Once the component’s belief values are continuously lower than 40%, the fault detection will be triggered and the system will create an alarm automatically. After the fault alarm is sent, the component’s belief value will be changed to almost 0 to avoid bad effects on data fusion and belief updating for other nodes.

Figure 11 indicate the results from the third experiment. It is obvious that the belief value of sensor 3 drops quickly after the sensor cannot work properly. Since the detected data are too different from the other two nodes, the belief value deceases quickly after a few times of the belief updating. The system can send an alarm about the self-detected fault.

Different from other reputation-based approaches, the proposed SBNCA can be used in the heterogeneous, large-scale IoT system because the whole system is divided into many smaller semantic match structures. Because any component only intends to observe the other components in its semantic match structure, the size and complexity of the entire IoT system will not affect the SBNCA much. For example, if we want to add more different types of sensors in several locations, we only need to add more semantic match structures to evaluate the behaviours of the newly added sensors without changing the existing system. For most of other approaches, involving a new type of sensor may require the overall redesign because it may totally change the system feature and behaviours.

## 6. Conclusions

In this paper, we propose a novel Semantic-based Belief Network Construction Approach for IoT systems to handle the uncertainty at runtime. The approach focuses on data accuracy with data fusion, and reliability with fault detection. Compared with some existing reputation-based framework for WSN applications [5], this framework can handle more heterogeneous systems via constructing the directed graphs with the semantic match mechanism. We implemented the approach in the IoTNetSim simulator with some extensions. Some experiments in the simulation environment are made to evaluate the proposed approach with the designed data structures and algorithms. In the experiments, the evaluation results indicate that the approach can work as expected.

The core structure of the proposed approach is built by the semantic match, which uses the semantics as the bridge to connect different physical devices and software components. The implementation of the semantic match discussed in this paper is rather simple. We only implemented and evaluated the location-aware semantics.

The future work is to enrich the expression of the semantic match with logic-based technologies to allow the framework to automatically detect the appropriate system components to construct the belief networks via the semantic match. Furthermore, we also plan to implement different algorithms to test the generality and extensibility of our approach.

## Figures and Tables

**Figure 1 sensors-20-05747-f001:**
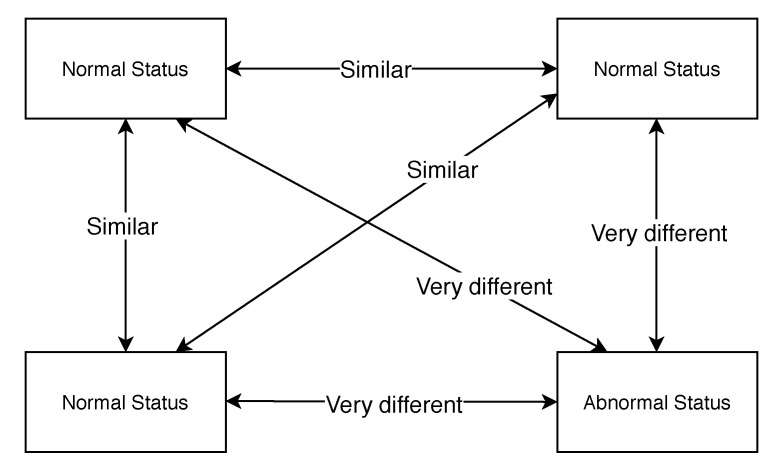
The assumption to use the components running normally to detect the faulty components with the same functionality.

**Figure 2 sensors-20-05747-f002:**
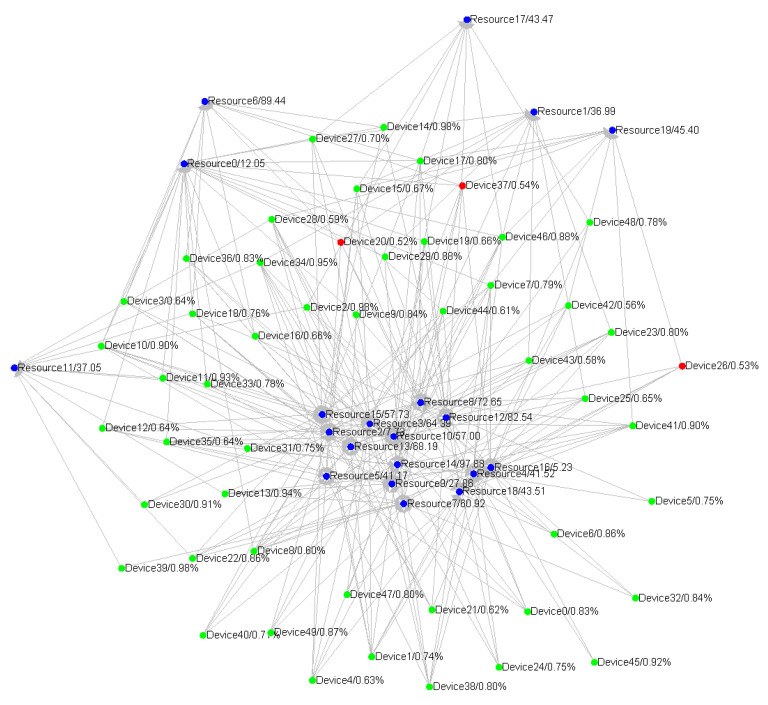
The visualised semantic-based belief network construction approach concept.

**Figure 3 sensors-20-05747-f003:**
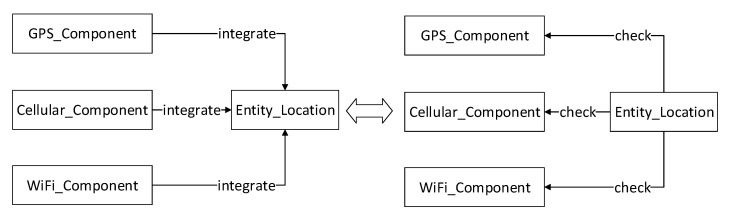
The hybrid localization solution as a semantic match example.

**Figure 4 sensors-20-05747-f004:**
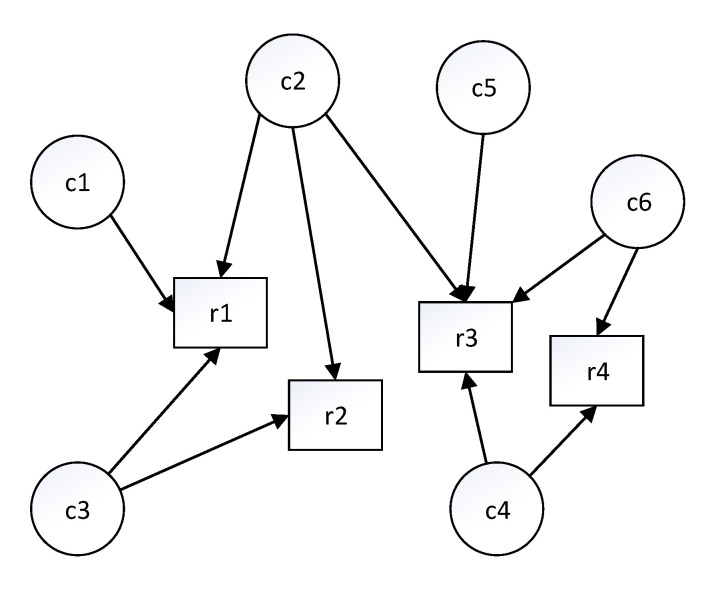
The semantic match is used to construct a directed graph.

**Figure 5 sensors-20-05747-f005:**
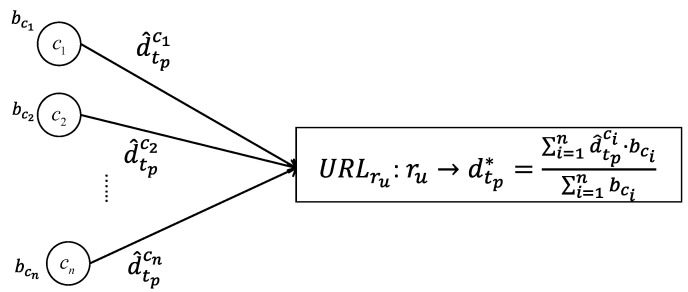
The data fusion process in the resource $r_{u}$ at the moment $t_{p}$.

**Figure 6 sensors-20-05747-f006:**
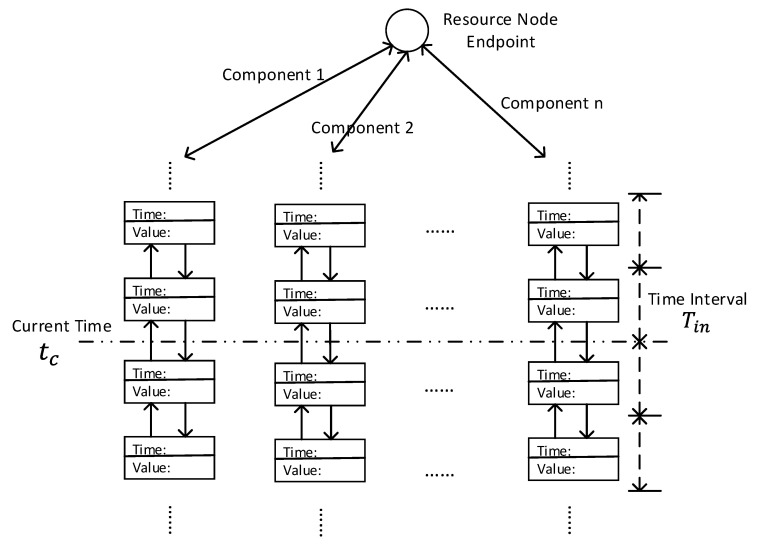
The data structure in the resource.

**Figure 7 sensors-20-05747-f007:**
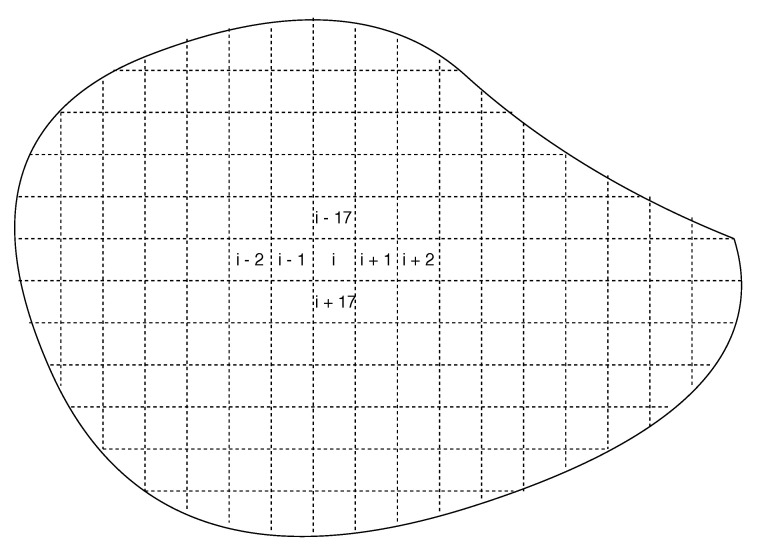
Construct the semantic match via the location-based grid in nature.

**Figure 8 sensors-20-05747-f008:**
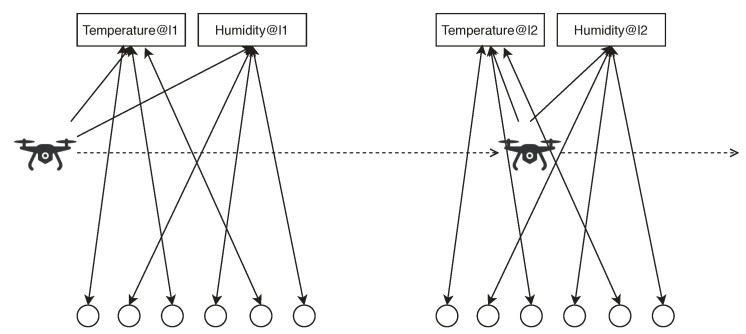
How the drone join and leave different semantic matches for data fusion, belief updating and fault detection in different areas.

**Figure 9 sensors-20-05747-f009:**
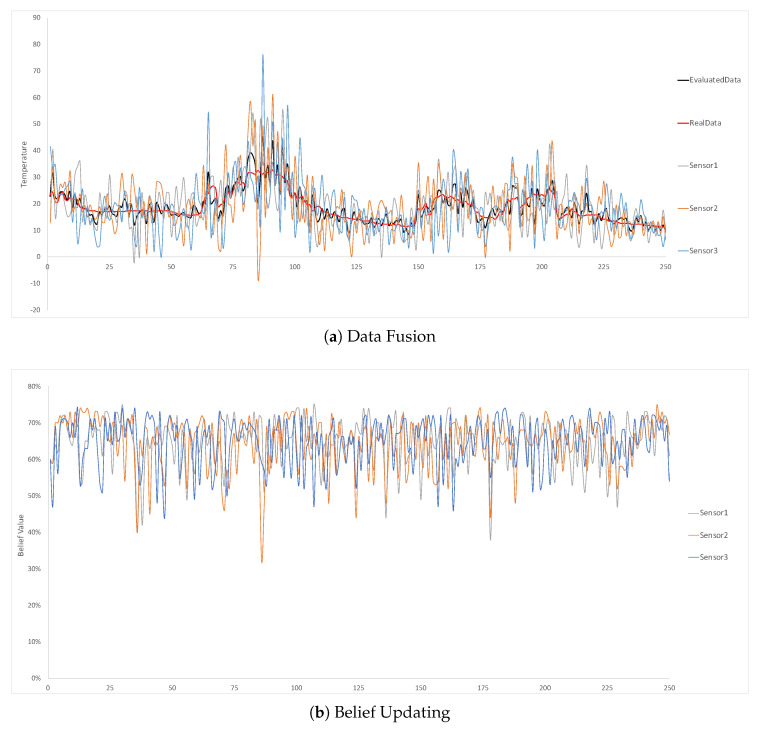
Three static temperature sensors at location 1 with 60% belief setting for each sensor.

**Figure 10 sensors-20-05747-f010:**
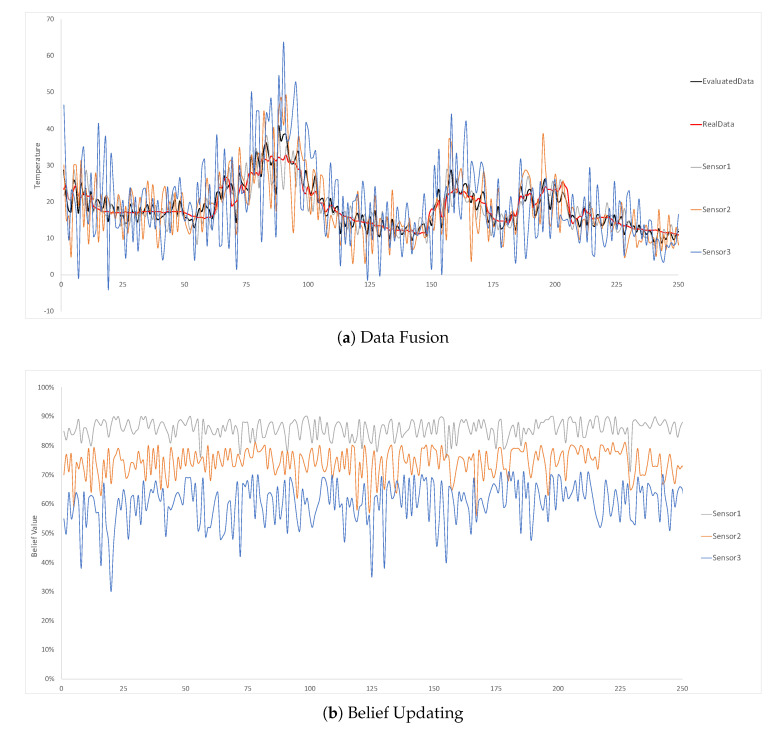
Three static temperature sensors at location 1 with 85%, 70% and 55% belief setting for each sensor, respectively.

**Figure 11 sensors-20-05747-f011:**
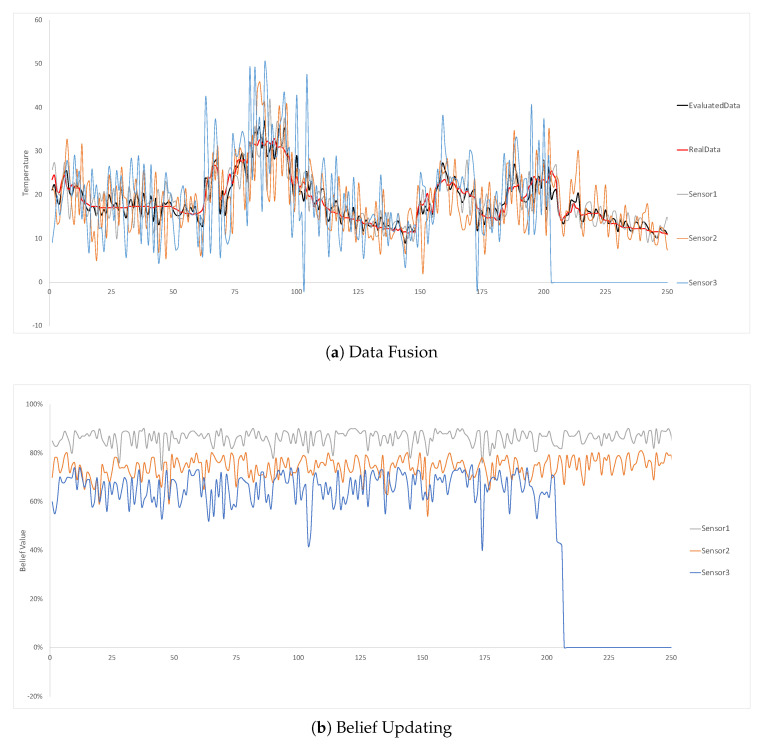
The permanent fault in sensor 3 is detected.

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
