# Peer review of "A Semantic-Based Belief Network Construction Approach in IoT"

_sensors, 2020, doi:10.3390/s20205747_

Round 1

Reviewer 1 Report

General comments:

The domain of IoT-based system monitoring including the parameters and the nature of sensors need to be clarified

The framework of the work with its current status is quite general and difficult to verity and follow. Some parts are corrected only with respect to one point and not with respect to the others!

The concentration of the work along with the target application needs to be defined!

The uncertainty and the physical environment must be clearly stated!

Assumption are required to be realistic!

Please clearly define the physical environment

The introduction is suffering from lack of literature survey!

Line 25: Please correct: we an use IoT systems to…

Line 52-53: Since the system components that point to

the same "resource" have the same type of functionality, they can evaluate each others’ behaviours and

 thus give a belief value. Please clarify: this is not necessarily correct! Different components pointing to the same source may evaluate different aspects of that particular source!

Line 62-66: I’m not clear on the concept and the evaluation of the assumptions that you have made, in the previous part and how you come to such points!

Please state your assumption properly with strong support as the whole system and the simulation is based on!

Please clearly state in Figure 1, what do you mean by faulty! Which aspects of the devices/nodes are faulty!

Line 127: In particular for device noise, the Data Fusion is used to support more

reliable and accurate values produced from the devices: how data fusion can affect the noise of signal which is correlated to the quality of the signal!!!

Line 134: Uncertainty of what? The IoT devices? the whole system?

I would strongly recommend defining the system and the devices in which you would include in the system. The nature and type of the sensors and output could quite different from one to another.

Line 144: the environment is the sensor with the device and related software component.

This is not the correct way of presenting the components in IoT!

Since I can not evaluate the concept of the manuscript with its current status, I would recommend to re-consider the structuring the manuscript with respect to terminologies, definitions, and assumptions!

Author Response

Dear Reviewer,

We've addressed all the mentioned points based on the comments and improved the paper. Thank you. 

Reviewer 2 Report

The proposed work is interesting, however, the description of it, the experimentations, and the assessments are rather simple.

Figure 1 is too complex to be in the Introduction. This figure is only used to explain the concept, why are there so many devices? A simple figure would be easier to understand and provide more aggregated value to the section. Additionally, this figure is first cited in line 55 and again in line 156, which are too far apart. The authors should avoid this type of citation. I think the figure can be used in Section 3. Maybe use a simpler one at the introduction.

The authors stated that “The whole belief updating mechanism is based on two assumptions which will be introduced in this paper with details.” (l.53) Which are? There is no space for suspense in scientific works, you have to give the spoilers right the way.

The related work should be improved. There are missing details about the works. Each one should have, at least, a brief description, the achieved results, the positive aspects of the work, as well as, the negative aspects (i.e., where the mentioned work can be improved).

Figure 2 does not add any important information to the work, so there is no need for it.

Figure 5 and 8 appears in the text before it is cited. All visual aids should be cited beforehand.

The experiments executed by the authors are very simple and their assessment is superficial. Additionally, the authors should compare the proposed approach with some of the related work.

The authors should not include new information in the Conclusion (line 428 to 432). These details should be discussed previously in the work.

Several missing commas. If you have 3 or more words, clauses, or phrases it is advised to separate them with commas. Please find below a few examples.
“challenges like scalability, inter-operability and fault tolerance” (l.18) and -> “interoperability”
“inaccurate detection, action or possible damage.” (l.34)
“such as physical randomness, noise, software faults or attack.” (l.74)
“such as scalability, lookup, communication protocols and social networking management” (l.107)

Some other typos and grammatical issues:
“we an use IoT systems” ( l.35)
“discussed in the respective Sections.” (l.130) -> section
“The Semantic Match is to construct” (l.130) -> is used to? Is adopted?
Double definition and citation (l.318) “PI (proportional-integral) control mechanism [36]” and (l.360) “PI
360 (Proportional Integral) control [38]”. Incidentally, why different citations for the same thing?

Author Response

Dear Reviewer,

We've addressed all the comments and improved the paper. Thank you.

Round 2

Reviewer 2 Report

I’m very pleased with the authors' comments and refinements to the manuscript.

There is only one point that a believed that could still be improved, the Related Work. The authors adjust some works there and add more information, but it still lacks the basics, as my previous comment points out: “The related work should be improved. There are missing details about the works. Each one should have, at least, a brief description, the achieved results, the positive aspects of the work, as well as, the negative aspects (i.e., where the mentioned work can be improved).”

Author Response

Dear Reviewer,

The related work section is improved based on your comments. There are still a few left citations are not given many details because they are literature review or summary papers.